# Endogenous Fungal Endophthalmitis: Causative Organisms, Treatments, and Visual Outcomes

**DOI:** 10.3390/jof8060641

**Published:** 2022-06-16

**Authors:** Kuan-Jen Chen, Ming-Hui Sun, Yen-Po Chen, Yi-Hsing Chen, Nan-Kai Wang, Laura Liu, An-Ning Chao, Wei-Chi Wu, Yih-Shiou Hwang, Chi-Chun Lai

**Affiliations:** 1Department of Ophthalmology, Chang Gung Memorial Hospital, Taoyuan 333, Taiwan; minghui0215@gmail.com (M.-H.S.); yenpo.chen@gmail.com (Y.-P.C.); yihsing@gmail.com (Y.-H.C.); lauraliu@gmail.com (L.L.); anningchao@hotmail.com (A.-N.C.); weichi666@gmail.com (W.-C.W.); yihshiou.hwang@gmail.com (Y.-S.H.); chichun.lai@gmail.com (C.-C.L.); 2College of Medicine, Chang Gung University, Taoyuan 333, Taiwan; 3Department of Ophthalmology, Tucheng Municipal Hospital, New Taipei 236, Taiwan; 4Department of Ophthalmology, Edward S. Harkness Eye Institute, Columbia University, New York, NY 10032, USA; wang.nankai@gmail.com; 5Department of Ophthalmology, Chang Gung Memorial Hospital, Keelung 204, Taiwan

**Keywords:** endogenous endophthalmitis, fungal endophthalmitis, intravitreal antibiotics, retinal detachment, vitrectomy

## Abstract

Endogenous fungal endophthalmitis (EFE) is a vision-threatening intraocular infection and a rare complication of fungemia. Early diagnosis and prompt aggressive treatment are crucial to avoid vision loss. We retrospectively reviewed the data of 37 patients (49 eyes) with EFE who were treated at a tertiary referral hospital from January 2000 to April 2019. The most common risk factor was diabetes (24 patients; 65%), followed by recent hospitalization, urinary tract disease, liver disease, and immunosuppressive therapy. Two or more risk factors were detected in 24 patients (65%), and yeasts (29 patients; 78%) were more commonly detected than mold (8 patients; 22%). The most common fungal isolates were *Candida* spp. (78%), especially *Candida albicans* (70%). Moreover, 24 eyes in 21 patients underwent vitrectomy, and 2 eyes underwent evisceration. Retinal detachment (RD) occurred in 17 eyes (35%) in 14 patients, and eyes without RD exhibited significantly superior visual outcomes (*p* = 0.001). A comparison of the initial VA between the better (20/200 or better) and worse groups (worse than 20/200) revealed that better initial VA was related to a superior visual outcome (*p* = 0.003). Therefore, to achieve superior visual outcomes, early diagnosis and prompt treatment are necessary for patients with EFE.

## 1. Introduction

Endogenous endophthalmitis, or metastatic endophthalmitis, is a vision-threatening intraocular infection and a rare complication of bacteremia or fungemia. In East Asian and Western countries, distinct causative organisms of endogenous endophthalmitis have been reported [1]. Gram-positive bacteria and *Candida* spp. are the most common pathogens in Western countries [1], whereas Gram-negative bacteria, especially *Klebsiella pneumoniae*, are the main causative organism in East Asian countries [2,3]. In East Asia, the incidence of endogenous fungal endophthalmitis (EFE) is relatively low; however, recent studies have reported an increase in EFE cases [4]. Risk factors for EFE in candidemia include an immunocompromised state, central venous catheter use, endocarditis, cirrhosis, diabetes with chronic complications, intravenous drug use (IVDU), radiation therapy, and solid organ transplantation [5,6,7]. A mortality rate of 5% to 71% in patients with systemic fungemia, specifically candidemia, suggests that infection progression into the eye may be a mortality indicator [8,9]. Early diagnosis and prompt aggressive treatments are crucial for avoiding vision loss in patients with EFE. We retrospectively reviewed the data of patients with EFE who were treated at a tertiary referral hospital in Taiwan.

## 2. Materials and Methods

This study is a retrospective case series of patients who received a diagnosis of EFE between January 2000 and April 2019 at a tertiary referral hospital in Northern Taiwan. The Institutional Review Board of Chang Gung Memorial Hospital in Taoyuan, Taiwan, approved the study protocol (IRB number: CGMH 201900614 B0 C601, 10 August 2019) and waived the requirement of obtaining written informed consent. All methods were performed in accordance with the relevant guidelines and regulations.

Patients were included if they had evidence of EFE in either eye, defined as the presence of anterior and posterior segment inflammation or characteristic fundus lesions on ophthalmic examination, and if one or more of the following was the case: a positive culture report obtained from aqueous, vitreous, blood, or other infectious site samples. Characteristic fundus lesions revealed yellowish fluffy chorioretinal infiltrates with ill-defined, irregular borders predominantly involving the posterior pole associated with varying degrees of vitritis. These discrete areas of vitritis may present with consolidated fungal abscess balls adjacent to one another in a “string of pearls” configuration in *Candida* endophthalmitis. Patients with EFE due to other causes, including postoperative (defined as EFE cases occurring less than 1 year after any eye surgery), corneal ulcer-related, surgery-related glaucoma filtration, or post-traumatic endophthalmitis, were excluded. The risk factors were defined as predisposing systemic conditions, including risk factors such as diabetes mellitus, recent hospitalization, liver disease, renal disease, respiratory disease, malignancy, indwelling lines, organ transplantation, HIV/AIDS, intravenous drug use, hyperalimentation, hemodialysis, immunosuppressive therapy, etc.

Data on sex, age, recent predisposing risk factors for infection, microbiological findings, clinical findings, treatment modalities, initial as well as final visual acuity (VA), and EFE recurrence were collected. All patients with EFE received intravenous antifungal agents, including amphotericin B, fluconazole, voriconazole, or caspofungin, prescribed by physicians soon after the diagnosis. Ophthalmologic consultation was provided during patients’ hospitalization upon physician’s request or during patients’ visit to the emergency or ophthalmology departments due to ocular symptoms or EFE suspicion. Fluconazole (50–100 μg/0.1 mL), voriconazole (50–100 μg/0.1 mL), or amphotericin B (5–10 μg/0.1 mL) was administered intravitreally in the eye with vitritis. A standardized management protocol for EFE was not established; instead, management decisions were made by individual physicians. Trans pars plana vitrectomy was performed in patients with uncontrollable infection and a poor clinical response to intravitreal and systemic antibiotic administration if patients’ systemic condition was suitable for undergoing vitrectomy. However, vitrectomy was not performed in eyes with gross structural and functional damage due to poor visual prognosis.

Poor visual outcome was defined as a VA worse than 20/200, whereas a favorable prognosis was defined as a VA of 20/200 or better. Statistical analyses were performed using the chi-square test or Fisher’s exact test. All analyses were performed using SPSS (version 26.0, IBM Corp., Armonk, NY, USA), and *p* ≤ 0.05 was considered statistically significant.

## 3. Results

### 3.1. Baseline Demographics and Clinical Characteristics

A summary of the demographics and clinical characteristics of patients is provided in Table 1. A total of 49 eyes in 37 patients, with an average age of 50.6 ± 27.6 years (range: 2–77 years), were included in the study. In the study population, 22 (59%) were men, and 15 were women. Moreover, 12 patients (32%) reported bilateral endophthalmitis, 14 (38%) had right eye involvement, and 11 (30%) had left eye involvement. Among patients with bilateral involvement, eight patients had similar lesions in both eyes at presentation, but two patients had obviously different severities of chorioretinal lesions and vitritis. The average follow-up duration was 14.8 months (range: 1–144 months; median: 6 months).

The interval between symptom onset and EFE presentation ranged from 3 to 60 days (mean: 15 days). However, the shorter interval (range 3–30 days; mean: 12 days) between symptom onset and EFE presentation was identified in patients with bilateral involvement. None of the patients were identified to have EFE through routine screening. Three patients, including two with an endotracheal tube and a 2-year-old boy, had no VA records at presentation. The most common ocular symptoms included decreased vision (41 eyes; 84%), redness (27 eyes; 55%), and pain (15 eyes; 31%). An initial diagnosis of EFE was conducted in 29 eyes (78%). The remaining patients were provided with a diagnosis of noninfectious uveitis or bacterial endophthalmitis. At initial evaluation, most eyes exhibited diffused anterior and posterior inflammation (43 eyes; 88%).

### 3.2. Microbiological Diagnosis

Table 2 lists the microbiological findings. Yeasts (29 patients; 78.4%) were more commonly detected than molds (8 patients; 21.6%). The most common fungal isolates were *Candida* spp. (78.4%), especially *Candida albicans* (70.3%). *Fusarium solani* and *Acremonium* spp. were predominant among molds. Moreover, 21 patients (57%) had a diagnosis of fungemia, and 19 patients (51%) had positive culture reports for intraocular fluid samples obtained through either tap or vitrectomy. Other positive cultures were obtained from urine, central venous pressure line, sputum, vagina, and esophagus samples.

### 3.3. Underlying Predisposing Conditions and Other Comorbidities

Most patients presented with at least one associated systemic medical condition (Table 3). The most common risk factor for EFE was diabetes mellitus (24 patients; 65%); 24 patients (65%) exhibited two or more risk factors. Moreover, 20 patients (54%) had been hospitalized over the past 6 months, and 4 patients died of multiorgan failure during admission.

### 3.4. Treatment and Visual Outcomes

Table 4 lists the treatment strategies and visual outcomes. All patients received systemic antifungal therapy with fluconazole, voriconazole, amphotericin B, or caspofungin. All patients, except one patient (patient 31), received intravitreal antifungal agents, including fluconazole, voriconazole, or amphotericin B. However, 24 eyes in 21 patients underwent vitrectomy, and 2 eyes in 2 patients underwent evisceration.

Retinal detachment (RD) occurred in 17 eyes (35%) in 14 patients, with primary RD occurring in 8 eyes and secondary RD in 9 eyes. The causative organisms were *C. albicans* (*n* = 7), *Aspergillus versicolor* (*n* = 1), *Acremoniums* spp. (*n* = 2), *Pseudallescheria boydii* (*n* = 1), *F. solani* (*n* = 1), *Cladosporium* spp. (*n* = 1), and *Phaeoacremonium* (*n* = 1). Seven eyes with primary RD did not undergo further surgical treatment because of the poor systemic condition and prognosis of the patients. Five eyes with secondary RD after vitrectomy did not undergo further surgical treatment because of poor prognosis. Of five eyes with primary (*n* = 1) or secondary (*n* = 4) RD, postoperative anatomic success was observed in three eyes. Compared with eyes with RD, eyes without RD exhibited significantly superior visual outcomes (*p* = 0.001).

Final VA data were available for 35 patients (45 eyes) at their last follow-up examination. For the remaining two patients, VA could not be assessed accurately because of limited mental status. A 2-year-old patient with *F. solani* endophthalmitis died during admission, and his vision was presumed to exhibit no light perception due to corneal perforation and phthisis bulbi. A comparison of initial and final VAs is provided in Table 5. Twenty-two eyes (44.9%), including 21 eyes with *Candida* infection and 1 eye with mold infection, achieved a VA of 20/200 or better. VA improved in 28 eyes (57%) after treatment, including in 15 eyes that underwent vitrectomy.

Eyes that underwent vitrectomy did not achieve better visual outcomes than eyes that did not undergo vitrectomy (*p* = 0.741). No significant difference in visual outcomes was observed among eyes treated with different antifungal agents. A comparison of initial VA between the better (20/200 or better) and worse groups (worse than 20/200) revealed that a better initial VA was associated with a superior visual outcome (*p* = 0.003). No patient reported EFE recurrence after therapy cessation; however, four patients died during hospitalization.

## 4. Discussion

For early diagnosis and prompt treatment of EFE, our study emphasizes the criticality of identifying the causal organisms, clinical features, risk factors, and visual outcomes. The most common organism causing EFE was *C. albicans*, and the most common clinical manifestation was vision loss. Diabetes mellitus was the most common risk factor for EFE, followed by recent hospitalization, urinary tract disease, liver disease, immunosuppressive therapy, and malignancy. Bilateral involvement was observed in 32% of patients with EFE, especially in those with *C. albicans* infection. Patients presenting with a better VA exhibited a superior visual outcome. Moreover, eyes without RD achieved a favorable visual outcome.

Early studies have reported EFE development in 0% to 37% of patients with candidemia [10]. A systematic review of approximately 7500 patients with EFE reported the presence of candidemia in less than 1% of patients [11,12]. A routine ophthalmologic consultation after the laboratory findings of systemic *Candida* septicemia is a low-value practice and is not recommended by the American Academy of Ophthalmology [11]. However, an ophthalmologic consultation is recommended for patients with signs or symptoms suggestive of ocular infection, regardless of *Candida* septicemia [11]. In this study, we did not identify any patients with EFE on screening examination conducted during hospitalization.

Many studies have identified various risk factors for EFE [5,6,7]. Identifying at-risk patients may preclude EFE in asymptomatic inpatients through an ophthalmologic consultation. The relative risk of EFE with candidemia is higher in immunocompromised patients (e.g., those undergoing radiation therapy), patients who had undergone solid organ transplantation, and patients with advanced diabetes, cirrhosis, corticosteroid use, IVDU, and endocarditis [5,6,7]. Five patients were presumed to be possible endogenous *Candida* endophthalmitis because they only had positive urine cultures and characteristic fundus lesions. Although the actual diagnosis for EFE is positive fungal cultures isolated from blood, eyes, and other samples, endogenous candida endophthalmitis may occur in patients with positive urine cultures and characteristic fundus lesions.

Chronic systemic conditions increase the risk of EFE in patients with candidemia. In our previous and current studies, diabetes was predominant among patients with bacterial endophthalmitis [2,3,13] and EFE. In a Taiwan-based study, Wang et al. [14] found diabetes mellitus in 8 (57%) of 14 patients with EFE. In another study, Chakrabarti et al. [15] reported uncontrolled diabetes in 5 (42%) of 12 patients with EFE. Uppuluri et al. [5] reported that cirrhosis and diabetes with chronic complications doubled the risk of EFE. However, other studies [6,7] have reported diabetes to be a less crucial risk factor for EFE. This may be attributed to the variations in the dietary habits and ethnicity of participants and limited case numbers. Essman et al. [16] reported that long-term intravenous catheter use (67%) was the major predisposing risk factor for EFE; moreover, they detected more than two risk factors in most patients. Lingappan et al. detected three or more risk factors in 24 of 51 patients, whereas we detected two or more risk factors in 24 patients (65%) in the present study.

As reported in the literature [17,18,19,20], patients with a history of IVDU exhibit a higher risk of EFE because of the direct inoculation of pathogens into the bloodstream. Mir et al. [21] reviewed 56,839 cases of endogenous endophthalmitis hospitalizations in the United States and reported that 13.7% had a history of IVDU. In a previous study, 9.8% and 1.8% of patients with IVDU had a diagnosis of *Candida* infection and disseminated candidiasis, respectively [21]. However, no statistical difference in *Aspergillus* infection was observed between patients with IVDU and patients without IVDU. In our study, four cases of *Candida* endophthalmitis were related to IVDU. Moreover, Paulus et al. [10] reported the presence of *Candida* endocarditis in 7.6% of patients with EFE. However, this study reported only one patient with IVDU and endocarditis.

Previous studies have reported *Candida* species, especially *C. albicans*, to be the most common pathogen causing EFE [4,6,16,20,22]. Our study also revealed that *C. albicans* was predominant in our patients in Taiwan. However, Das et al. [23], in a recent study, reported *Aspergillus* spp. to be the most common pathogen in patients with EFE, followed by *Candida* and *Fusarium* spp. This variation may be attributed to geographic and racial differences.

The management of patients with EFE includes systemic antifungal antibiotics, intravitreal antifungal agents, and vitrectomy. However, the definitive role of early vitrectomy for candida endophthalmitis treatment was not established. Although some studies [4,15] have recommended vitrectomy for patients with EFE at presentation, other studies [24] have reported mixed results regarding its benefit on visual outcomes. Sallam et al. [24] suggested that early vitrectomy may be effective in reducing the risk of RD and might, therefore, be considered an initial treatment for fungal lesions penetrating the vitreous cavity. However, some studies [22,25] have reported that RD occurred after vitrectomy, especially in eyes with large chorioretinal infiltrate areas. In a case series, Lingappan et al. [22] reported a 29% (7 eyes) incidence rate of RD. In the seven eyes, RD developed 1 month after vitrectomy, suggesting peripheral vitreous contraction and consequent retinal break as a cause [22]. Chen et al. [25] suggested that chorioretinal lesions with adjacent vitreous infiltration have significant adhesion with the vitreous, retina, and choroid. In our study, RD occurred in 17 eyes (35%) in 14 patients, with primary RD occurring in 8 eyes and secondary RD in 9 eyes. We also observed a higher incidence of RD after vitrectomy in the eyes of patients with EFE. Therefore, we propose that the contraction of the vitreous–retina–choroid infiltrative complex in EFE contributes to tractional RD development, especially in a large choroidal infiltrate area.

Takebayashi et al. [26] assessed the association between EFE severity and visual prognosis and reported that all patients with early-stage EFE reported a final VA better than 20/200; among them, 40% reported a final VA better than 20/32. However, six eyes of patients with advanced-stage EFE had no light perception [26]. Sallam et al. reported the presence of centrally located fungal lesions in 44 eyes in 36 patients with EFE [24] and suggested that poor visual outcome is also related to poor initial VA. Therefore, poor initial VA and central retinal lesions are the most crucial risk factors for poor visual prognosis. Lipgappan et al. reported a VA of 20/200 or better in 28 (56%) eyes with yeast infection and in 5 (33%) eyes with mold infection. In our study, 22 eyes (45%) achieved a VA of 20/200 or better, including 21 eyes with *Candida* infection and 1 eye with mold infection. A superior visual outcome (*p* = 0.003) was reported in patients with a better initial VA. Among patients with a VA of 20/200 or better, most had a diagnosis of early-stage EFE and a noncentrally located fungal lesion. Compared with eyes with RD, eyes without RD exhibited significantly superior visual outcomes (*p* = 0.001).

The major limitations of our study include its retrospective design, the lack of a uniform protocol for diagnosis or treatment, and limited and variable follow-ups. All of these limited our ability to analyze the factors that may influence the visual outcome. Despite these drawbacks, the results of this case series confirm previous findings of a predominance of *Candida* spp. in EFE cases and poor visual outcomes among mold cases. In addition, this study confirmed the high incidence of bilateral involvement in *Candida* endophthalmitis and documented a final VA of 20/200 or better in 45% of eyes with EFE.

## 5. Conclusions

Our study demonstrated that approximately half of the patients with EFE exhibited favorable visual outcomes after treatment. Crucial risk factors for EFE include diabetes mellitus, recent hospitalization, urinary tract diseases, liver diseases, immunosuppressive therapy, and malignancy. Moreover, poor initial VA and RD development were associated with poor visual outcomes. Therefore, to achieve superior visual outcomes, early diagnosis and prompt treatment are necessary for patients with EFE.

## Figures and Tables

**Table 1 jof-08-00641-t001:** Demographics of patients with endogenous fungal endophthalmitis.

No. of	Sex/Age/Eye	Pathogen	Infectious Source or	DM	Cultures			Other Conditions
Patient			Major Disease		Blood	Eye	Others	
1	M/64/OU	*Candida albicans*	UTI				urine	nephrotic syndrome, pneumoconiosis, old TB
2	F/56/OU	*Candida albicans*	UTI	+			urine	renal stone, HD
3	M/52/OD	*Candida albicans*	UTI	+			urine	HBV, liver cirrhosis
4	F/37/OD	*Candida albicans*	UTI	+			urine	renal stone, CVA, CRI, candida osteomyelitis, HCV
5	F/77/OD	*Candida albicans*	undetermined	+		+	vagina	vulvovaginal candidiasis
6	M/45/OD	*Candida albicans*	esophageal cancer	+			esophagus	alcoholic liver cirrhosis, IMM
7	F/28/OD	*Candida albicans*	IVDU		+	+	urine	Heroin dependence, HIV, HBV, HCV
8	M/55/OS	*Candida albicans*	undetermined	+				old TB
9	F/55/OD	*Candida albicans*	UTI (APN)	+			urine	vaginal candidiasis, APN, renal stone
10	M/36/OS	*Candida albicans*	IVDU		+		urine	HBV, HCV
11	M/58/OD	*Cladosporium* spp.	myelodysplastic syndrome		+	+		IMM
12	M/58/OU	*Candida tropicalis*	fungemia	+	+		CVP line	pneumonia, pancreatitis with abscess, AMI
13	M/47/OU	*Candida albicans*	UTI (APN)	+	+		urine	HCV
14	M/34/OU	*Candida albicans*	UTI	+			urine	hydronephrosis with stone, HBV, DM
15	M/59/OU	*Candida albicans*	rectal cancer with metastasis		+			pneumoniae, HBV, HCV, lung metastasis, IMM
16	F/36/OD	*Candida albicans*	postpartum infection		+			fasciitis
17	F/70/OD	*Acremoniums* spp.	undetermined	+		+		CKD, HD, CHF
18	M/47/OS	*Candida albicans*	undetermined	+		+		chronic pancreatitis
19	M/62/OD	*Pseudallescheria boydii*	undetermined	+		+		CRI, pneumonia, septic shock
20	M/57/OS	*Aspergillus versicolor*	undetermined	+		+		HCV, liver cirrhosis, ascites
21	F/21/OS	*Candida parapsilosis*	dilation and curettage					
22	F/57/OU	*Candida albicans*	ovarian cancer	+	+			IMM
23	M/58/OU	*Candida albicans*	hepatoma	+	+			HBV, HCV, liver cirrhosis, IMM
24	M/58/OU	*Candida albicans*	fungemia	+	+	+		COPD; Salmonella enterica serogroup D, liver abscess
25	F/38/OS	*Candida parapsilosis*	fungemia	+	+	+		hyperthyroidism, liver abscess
26	M/2/OS	*Fusarium solani*	β-thalassemia major with CBT		+	+		IMM
27	M/44/OD	*Acremoniums* spp.	undetermined	+		+		pneumonia
28	F/58/OS	*Phaeoacremonium*	undetermined	+		+		
29	F/66/OD	*Candida albicans*	IVDU	+	+			Heroin dependence, forearm cellulitis, CAD
30	M/41/OD	*Candida albicans*	HCC with metastasis		+	+		HBV, HCC, liver cirrhosis, IMM
31	F/50/OU	*Candida albicans*	ovarian cancer		+			IMM
32	F/73/OS	*Candida albicans*	undetermined		+	+		CKD
33	M/54/OS	*Fusarium solani*	AML	+	+	+		IMM
34	F/24/OD	*Candida albicans*	dilation and curettage		+	+		
35	M/63/OU	*Candida albicans*	rectal cancer with metastasis		+	+		IMM
36	M/34/OS	*Candida albicans*	ESRD with renal transplantation	+	+		urine	IgA nephropathy, renal transplantation, IMM
37	M/38/OU	*Candida albicans*	IVDU, IE	+	+	+		HF, pulmonary emboli, mediastinal abscess

AMI; acute myocardial infarction; AML, acute myeloid leukemia; APN, acute pyelonephritis; CAD, cardiovascular disease; CBT, cord blood transplantation; CKD, cystic kidney disease; COPD, chronic obstructive pulmonary disease; CRI: chronic renal insufficiency; CVP: central venous pressure; ESRD, end-stage renal disease; HBV, hepatitis B virus; HCC, hepatocellular carcinoma; HCV, hepatitis C virus; HD, hemodialysis; HF, heart failure; IE, infective endocarditis; IMM, immunosuppressants; IVDU, intravenous drug use; TB, tuberculosis; UTI: urinary tract infection.

**Table 2 jof-08-00641-t002:** Pathogens of Exogenous Fungal Endophthalmitis.

Pathogens	No. of Patients	Percent
*Candida albicans*	26	70.3%
*Candida parapsilosis*	2	5.4%
*Candida tropicalis*	1	2.7%
*Fusarium solani*	2	5.4%
*Acremonium* spp.	2	5.4%
*Aspergillus versicolor*	1	2.7%
*Cladosporium* spp.	2	2.7%
*Phaeoacremonium* spp.	1	2.7%
*Pseudallescheria boydii*	1	2.7%
Total	37	100.0%

**Table 3 jof-08-00641-t003:** Risk factors of endogenous fungal endophthalmitis.

Risk Factor *	Number of Patients	Percent
Diabetes mellitus	24	64.8%
Recent hospitalization (within 6 months)	20	54.1%
Urinary tract disease	11	29.7%
Liver disease	11	29.7%
Immunosuppressive therapy	11	29.7%
Malignancy	8	21.6%
Intravenous line	6	16.2%
Indwelling urinary catheter	6	16.2%
Respiratory disease	5	13.5%
Intravenous drug use	4	10.8%
Vagina-related diseases	4	10.8%
Total parenteral nutrition	3	8.1%
Hematologic diseases	3	8.1%
Hemodialysis	2	5.4%
Organ transplantation	2	5.4%
Tuberculosis	2	5.4%
Pancreatitis	2	5.4%
Cardiac diseases	2	5.4%
HIV infection	1	2.7%

* Twenty-four patients had 2 or more risk factors.

**Table 4 jof-08-00641-t004:** Visual outcomes and treatment in patients with exogenous fungal endophthalmitis.

No. of	Sex/Age/Eye	Pathogen	Initial VA	Treatment		RD	Final VA	Cause of	Follow-Up
Patient				Vitrectomy	Antifungal Agents			Poor VA	(Months)
1	M/64/OU	*Candida albicans*	0.06/0.2	+/-	Amp	-/-	0.2/0.6		7
2	F/56/OU	*Candida albicans*	CF/CF	+/+	Amp	+/+	HM/HM	RD/RD	4
3	M/52/OD	*Candida albicans*	0.3	+	Amp	-	0.5		6
4	F/37/OD	*Candida albicans*	CF	+	Flu	-	0.02		45
5	F/77/OD	*Candida albicans*	LP	+	Flu	+	NLP	evisceration	1
6	M/45/OD	*Candida albicans*	CF	+	Flu	-	0.05		17
7	F/28/OD	*Candida albicans*	CF	+	Vor	-	0.1		5
8	M/55/OS	*Candida albicans*	HM	-	Flu	-	0.1		12
9	F/55/OD	*Candida albicans*	0.03	+	Vor	-	0.4		15
10	M/36/OS	*Candida albicans*	CF	+	Vor	-	0.2		144
11	M/58/OD	*Cladosporium* spp.	CF	+	Amp, Vor	+	0.1		81
12	M/58/OU	*Candida tropicalis*	NA	-	Vor	-	NA (expired)		2
13	M/47/OU	*Candida albicans*	0.6/0.05	-/+	Vor	-/-	1.0/0.4		6
14	M/34/OU	*Candida albicans*	0.3/CF	-/+	Vor	-/-	1.0/CF	macular scar (os)	8
15	M/59/OU	*Candida albicans*	NA	-	Flu	-	NA (expired)		2
16	F/36/OD	*Candida albicans*	0.02	+	Vor, Amp	-	0.5		6
17	F/70/OD	*Acremoniums* spp.	HM	-	Amp	+	NLP	phthisis	3
18	M/47/OS	*Candida albicans*	NLP	-	Flu	+	NLP	evisceration	1
19	M/62/OD	*Pseudallescheria boydii*	CF	-	Vor	+	HM	RD	2
20	M/57/OS	*Aspergillus versicolor*	NLP	-	Amp	+	NLP	phthisis	2
21	F/21/OS	*Candida parapsilosis*	CF	-	Flu	-	0.1		5
22	F/57/OU	*Candida albicans*	0.3/0.3	-/-	Flu	-/-	1.0/1.0		18
23	M/58/OU	*Candida albicans*	CF/0.01	-/-	Flu	-/-	0.05/0.08 (expired)		2
24	M/58/OU	*Candida albicans*	0.01/CF	+/+	Flu	-/+	0.06/LP	phthisis (os)	8
25	F/38/OS	*Candida parapsilosis*	0.2	-	Flu	-	1.0		58
26	M/2/OS	*Fusarium solani*	NA	-	Flu	+	NLP (expired)		2
27	M/44/OD	*Acremoniums* spp.	LP	-	Flu	+	NLP	phthisis	2
28	F/58/OS	*Phaeoacremonium* spp.	HM	+	Flu, Amp	+	NLP	phthisis	2
29	F/66/OD	*Candida albicans*	0.05	-	Flu	-	0.4		7
30	M/41/OD	*Candida albicans*	0.1	+	Vor	-	0.2		7
31	F/50/OU	*Candida albicans*	0.5/0.6	-/-	Flu	-/-	1.0/1.0		35
32	F/73/OS	*Candida albicans*	0.1	+	Flu	+	LP	RD	3
33	M/54/OS	*Fusarium solani*	HM	+	Vor, Amp	-	LP	corneal opacity	7
34	F/24/OD	*Candida albicans*	LP	+	Amp	-	0.8		6
35	M/63/OU	*Candida albicans*	0.02/CF	+/+	Flu, Vor, Amp	+/+	0.2/HM	RD (os)	12
36	M/34/OS	*Candida albicans*	0.01	-	Flu	-	0.01		2
37	M/38/OU	*Candida albicans*	NLP/CF	-/+	Flu	+/+	NLP/0.2	RD (od)	8

Amp, amphotericin B; CF, counting fingers; Flu, fluconazole; HM, hand motions; LP, light perception; NLP, no light perception; RD, retinal detachment; VA, visual acuity; Vor, voriconazole.

**Table 5 jof-08-00641-t005:** Visual outcomes of eyes with endogenous fungal endophthalmitis.

	Initial Visual Acuity		Final Visual Acuity	
	No. of Eyes	Percent	No. of Eyes	Percent
>20/40	3	6.1%	11	22.4%
20/200–20/50	8	16.3%	11	22.4%
2/200–19/200	9	18.4%	9	18.4%
Counting fingers	14	28.6%	1	2.0%
Hand motions	4	8.2%	4	8.2%
Light perception	3	6.1%	3	6.1%
No light perception	3	6.1%	7	14.3%
Not available	5	10.2%	4	8.2%
Total	49	100%	49	100%

## Data Availability

The data analyzed during this study are available upon request from the corresponding author: Kuan-Jen Chen. The data are not publicly available due to containing information that could compromise the privacy of the research participants.

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
