# Peer review of "Endogenous Fungal Endophthalmitis: Causative Organisms, Treatments, and Visual Outcomes"

_jof, 2022, doi:10.3390/jof8060641_

Round 1
Reviewer 1 Report
‘Endogenous Fungal Endophthalmitis: Causative Organisms, Treatments, and Visual Outcomes’ is a hospital-based description of endogenous fungal endophthalmitis treated over several years.
This is good data, but its value would increase only if the authors perform a statistical analysis to identify the risk factors of EFE.
Also, I will suggest
Line 62- Briefly mention the ‘characteristic fundus lesions’ in EFE
Line 80- Surprisingly, the authors performed pars plana vitrectomy only in selected patients- ‘only for uncontrolled infection’. Several studies suggest a superior visual and structural outcome after vitrectomy in fungal endophthalmitis.
Table 1- I suggest adding ‘percentage’ to the absolute number to make it more meaningful
Finally, the manuscript needs extensive language and grammar corrections.
Author Response
‘Endogenous Fungal Endophthalmitis: Causative Organisms, Treatments, and Visual Outcomes’ is a hospital-based description of endogenous fungal endophthalmitis treated over several years.
This is good data, but its value would increase only if the authors perform a statistical analysis to identify the risk factors of EFE.
Ans: Thank you for your comments. We agree that the statistical analysis to identify the risk factors of EFE is beneficial for patients with fungemia. Because this study is not to identify the risk factors from patients with fungemia/candidemia, we cannot actually identify the risk factors. We only can find the risk factors in patients with EFE.
Also, I will suggest
Line 62- Briefly mention the ‘characteristic fundus lesions’ in EFE
Ans: Thank you for your comments. We add the he ‘characteristic fundus lesions’ in EFE. Characteristic fundus lesions revealed yellowish fluffy chorioretinal infiltrates with ill-defined, irregular borders predominantly involving the posterior pole associated with varying degrees of vitritis. These discrete areas of vitritis may present with consolidated fungal abscess balls adjacent to one another in a "string of pearls" configuration in Candida endophthalmitis.
Line 80- Surprisingly, the authors performed pars plana vitrectomy only in selected patients- ‘only for uncontrolled infection’. Several studies suggest a superior visual and structural outcome after vitrectomy in fungal endophthalmitis.
Ans: Thank you for your comments. Most patients did not undergo vitrectomy because these patients had no light perception/light perception vision or poor systemic condition. After discussion with patients and their family, all these patients did not agree to undergo vitrectomy. In discussion section, we discuss this issue of vitrectomy.
Table 1- I suggest adding ‘percentage’ to the absolute number to make it more meaningful.
Ans: Thank you for your comments. There is no need of percentage in Table 1 We presume the adding data in Table 3. We add percentage in the table 3.
Finally, the manuscript needs extensive language and grammar corrections.
Ans: Thank you for your comments. This manuscript was edited by Wallace Academic Editing and provided a certificate for extensive language and grammar corrections.
Reviewer 2 Report
The authors present a retrospective of a series of endophthalmitis cases followed in a Twain hospital between 2000 and 2019. They discuss clinical presentation, isolated infectious agents, risk factors, treatment and clinical outcomes.
The number of publications on the subject has increased over the years, but the rarity and difficulty of managing patients make information important. The data is consistent and well-organized, and the authors point out limitations due to the study design. This does not prevent the demonstration of data. Some points could have been more detailed:
1- The risk factor diabetes mellitus (decompensated? how severe? was there any blood glucose value above which the risk increased? patients with chronic decompensation? etc.) Is it possible to check glicemia levels in order to indicate critical values?;
2- Make more clear that the immunocompromised risk factor is transversal to all related risk factors (negative regulation of the immune response ? most of risk factor are linked to decrease of immunity when not controlled);
3- Does bilateral involvement worsen the prognosis? When cases of bilateral involvement was detected, were the lesions similar?
4- Why racial differences would influence the disease outcome?
Author Response
The authors present a retrospective of a series of endophthalmitis cases followed in a Twain hospital between 2000 and 2019. They discuss clinical presentation, isolated infectious agents, risk factors, treatment and clinical outcomes.
The number of publications on the subject has increased over the years, but the rarity and difficulty of managing patients make information important. The data is consistent and well-organized, and the authors point out limitations due to the study design. This does not prevent the demonstration of data. Some points could have been more detailed:
- The risk factor diabetes mellitus (decompensated? how severe? was there any blood glucose value above which the risk increased? patients with chronic decompensation? etc.) Is it possible to check glicemia levels in order to indicate critical values?;
Ans: Thank you for your comments. Glycemia were highly variable between individuals in our study. We did not find any association between blood sugar level and EFE. Therefore, we did not provide this information.
- Make more clear that the immunocompromised risk factor is transversal to all related risk factors (negative regulation of the immune response ? most of risk factor are linked to decrease of immunity when not controlled);
Ans: We agree this comment. Immunocompromised status is considered the most important risk factor of EFE; therefore, we investigated the risk factors in Immunocompromised patients of our study.
In method section, we add the immunocompromised risk factors as follows.
The immunocompromised risk factors are defined as predisposing systemic conditions, including risk factors such as diabetes mellitus, recent hospitalization, liver disease, renal disease, respiratory disease, malignancy, indwelling lines, organ transplantation, HIV/AIDS, intravenous drug use, hyperalimentation, hemodialysis, immunosuppressive therapy, etc.
3- Does bilateral involvement worsen the prognosis? When cases of bilateral involvement was detected, were the lesions similar?
Ans: Thank you for your comments.
Bilateral involvement doesn’t worsen the prognosis. Eleven eyes (55%) of 20 eyes had a 20/200 vision or better.
Eight patients had similar lesions at presentation, but two patients had obviously different severities of choriioretinal lesions and vitritis. The shorter interval (range 3-30 days; mean: 12 days) between symptom onset and EFE presentation was identified in patients with bilateral involvement.
We add these data in the result section.
4- Why racial differences would influence the disease outcome?
Ans: Thank you for your comments. The visual outcome of EFE depends on early diagnosis and prompt treatment. The racial differences influence the presentations and varieties of types of organisms. We did not mention that racial differences would influence the disease outcome.
Round 2
Reviewer 1 Report
Please see the attched file

Author Response
Dear Reviewer,
Thank you for your concerns and consideration. We appreciate the review’s comments and corrections. We try our best to revise our manuscript. We hope this revision can met the criteria of journal. Please find the attached file of revised manuscript.
Sincerely yours,
Kuan-Jen Chen, MD
